# Atypical Foveal Hypoplasia in Best Disease

**DOI:** 10.3390/jpm13020337

**Published:** 2023-02-15

**Authors:** Emmanuelle Moret, Raphaël Lejoyeux, Sophie Bonnin, Georges Azar, Jessica Guillaume, Chloé Le Cossec, Justine Lafolie, Anne-Sophie Alonso, Catherine Favard, Isabelle Meunier, Vivien Vasseur, Martine Mauget-Faÿsse

**Affiliations:** 1Plateforme d’Investigation Clinique, Hôpital Fondation Adolphe de Rothschild, 75019 Paris, France; 2Unité de Recherche Clinique, Hôpital Fondation Adolphe de Rothschild, 75019 Paris, France; 3Centre Ophtalmologique de l’Odéon, 113 Boulevard Saint Germain, 75006 Paris, France; 4CHU Gui de Chauliac, 80 Avenue Augustin Fliche, 34090 Montpellier, France

**Keywords:** Best disease, fovea plana, foveal hypoplasia, foveal avascular zone, OCT-A

## Abstract

Purpose: To determine the prevalence and characteristics of foveal hypoplasia (also called fovea plana) in patients with Best disease using spectral-domain (SD) optical coherence tomography (OCT) and OCT-angiography (OCT-A). Design: A retrospective observational study including patients diagnosed with Best disease. Subjects and Participants: Fifty-nine eyes of thirty-two patients (fifteen females (46.9%) and seventeen males (53.1%), *p* = 0.9) diagnosed with Best disease were included. Patients’ eyes were categorized into two groups: Eyes with a fovea plana appearance (‘FP group’) and eyes without fovea plana appearance (‘no FP group’), based on the foveal appearance on B-scan SD-OCT. Methods and Main Outcome Measures: Cross-sectional OCT images were assessed for the persistence of inner retinal layers (IRL) and OCT-A was analyzed for the presence of a foveal avascular zone (FAZ), the size of which was determined when applicable. Results: Overall, 16 eyes (27.1%) of 9 patients had a fovea plana appearance (‘FP group’) with the persistence of IRL, and 43 eyes (72.9%) of 23 patients did not have fovea plana appearance (‘no FP group’). Among FP eyes, OCT-A performed in 13 eyes showed bridging vessels through the FAZ in 100% of eyes with OCT-A. Using Thomas classification, 14 out of the 16 eyes with fovea plana (87.5%) had atypical foveal hypoplasia, and the 2 others (12.5%) had a grade 1b fovea plana. Conclusion: In our series, foveal hypoplasia was present in 27.1% of patients with Best disease. OCT-A showed bridging vessels through the FAZ in all eyes. These findings highlight the microvascular changes associated with Best disease, which can be an early sign of the disease in patients with a family history.

## 1. Introduction

Best vitelliform macular dystrophy (BVMD), also called Best disease, is a rare and slowly progressive macular dystrophy with an autosomal dominant inheritance pattern in the majority of cases. It is characterized by the accumulation of subretinal lipofuscin deposits at the macula [1]. The disease is due to a mutation in the VMD2 or BEST1 gene on chromosome 11q12-q13 (OMIM #607854), which codes for the Bestrophin1 protein, a calcium-activated chloride channel located at the basolateral plasma membrane of retinal pigment epithelium (RPE) cells [2,3,4].

Clinical manifestations of BVMD vary depending on the stage of the disease. Six clinical stages are considered: Stage 1 (previtelliform), where only subtle RPE changes are present; stage 2 (vitelliform), with the characteristic bilateral yellow “egg-yolk” appearance of the macula; stage 3 (pseudohypopion), where some of the yellow matter that causes the egg yolk-like blister can breakthrough a layer under the retina; stage 4 (vitelliruptive), where the breakup of the material gives a “scrambled egg” appearance; stage 5 (atrophic), when central RPE and retinal atrophy appears; and, finally, stage 6 (choroidal neovascularization, CNV), with the development of CNV on or under the macula [5].

Best disease is diagnosed based on clinical appearance, a reduced light peak–dark trough (Arden) ratio (less than 170%) on an electrooculogram (EOG) [6,7], and with genetic testing. Fundus autofluorescence (FAF) may also confirm the presence of lipofuscin deposits, which typically exhibit intense autofluorescence [8].

Spectral domain (SD)-optical coherence tomography (OCT) and optical coherence tomography-angiography (OCT-A) have shown great promise for microstructure visualization and noninvasive mapping of retinal vasculature at the microcirculation level, allowing a better understanding of macular development and diseases. In Best disease, a lipofuscin deposit is visualized on B-scan SD-OCT within the RPE, the subretinal space, and the photoreceptors’ zone. In the subclinical stage, the thickening of a highly reflective band between the RPE/Bruch’s membrane complex and the ellipsoid zone (EZ)—attributable to unsheathed cone outer segments in the RPE—has been described [9]. As the disease progresses, a disintegration of the overlying EZ takes place, followed by the digestion of the material and further disintegration of the surrounding RPE/Bruch’s membrane complex, EZ, and photoreceptors. In the end stage, atrophic lesions with loss of the outer retina appear. In the case of CNV, subretinal/intraretinal fluid and hemorrhages are observed [5].

In recent reports, the use of OCT-A on eyes with Best disease revealed abnormalities in the foveal avascular zone (FAZ). Guduru et al. demonstrated patchy vascularity loss in the superficial and deep layers of the retina, as well as a hyporeflective center in the choriocapillaris layer, which represents a capillary dropout for these authors. Furthermore, they showed an enlarged FAZ area compared to healthy eyes [10]. On the other hand, Mirshahi et al. described the presence of bridging vessels in the FAZ of four eyes with BVMD [11], suggesting a fovea plana appearance.

Fovea plana, also called foveal hypoplasia [12], is defined as the persistence of inner retinal layers (IRL) at the fovea on OCT B-scans and the absence of FAZ on OCT-A [11,13,14,15,16]. This particular appearance of the fovea can be associated with conditions such as albinism, PAX6 mutation, aniridia, retinopathy of prematurity (ROP), SLC38A8 mutations, optic nerve hypoplasia, achromatopsia, and cone-rod dystrophy, but can also be found in normal children [14,17]. Using SD-OCT images, Thomas et al. proposed a structural grading system of foveal hypoplasia: Grade 1a/1b shows a shallow foveal pit, outer nuclear layer (ONL) widening, and outer segment (OS) lengthening relative to the parafoveal ONL and OS length, respectively; grade 2 has all the features of grade 1 without a foveal pit; grade 3 is the same as grade 2 except for the widening of the cones’ OS; grade 4 has all the features of grade 3 except no widening of ONL at the fovea; and grade atypical is characterized by a shallower pit with disruption of the inner EZ [14,18].

The purpose of this study is to analyze the fovea in patients presenting with Best disease thanks to B-scan OCT and OCT-A images, looking particularly for the presence of fovea plana and bridging vessels in the FAZ, and to discuss possible underlying mechanisms of these abnormalities.

## 2. Materials and Methods

### 2.1. Study Design

This is an observational, retrospective, monocentric study. The medical records of 33 patients (66 eyes) diagnosed with Best disease and followed at the Rothschild Foundation Hospital, Paris, France, between January 2013 and January 2022, were reviewed. Approval was obtained from the institutional review board (IRB 00012801) and the Ethics Committee (No. IDRCB IRB00012801) of the Rothschild Foundation Hospital, Paris—under the study number CE_20211123_7_MMT. The study adhered to the tenets of the Declaration of Helsinki and the International Conference of Harmonization Good Clinical Practice guidelines.

### 2.2. Participants and Data Collection

Demographic and medical data of included patients were extracted from medical files. All patients underwent a standard ophthalmologic examination including best-corrected visual acuity (BCVA) measurements, slit lamp, and fundus examination. BCVA was converted to the logMAR scale for statistical purposes. For each patient, available color fundus photography, FAF (Optos Panoramic 200Tx imaging system, Optos PLC, Dunfermline, Scotland, UK), and SD-OCT B-scans (Spectralis^®^ HRA-OCT device, Heidelberg Engineering, Heidelberg, Germany) with enhanced depth imaging (EDI) and/or OCT-A modules were analyzed. To be included, all patients should have had macular OCT imaging. Eyes were excluded if the quality of the OCT did not allow image data analysis.

The diagnosis of Best disease was made based on family history, clinical examination, and a reduced light peak–dark trough (Arden) ratio on EOG. An Arden ratio of less than 170% was defined as abnormal (Best disease confirmed), and a ratio between 170% and 179% was defined as borderline (highly suggestive of Best disease). EOG was performed according to the protocol approved by the International Society for Clinical Electrophysiology of Vision [7]—or genetic confirmation of a mutation in the BEST1 gene. Best disease stage (stages 1 to 6) and persistence of vitelliform material under the macula were retrospectively established based on fundus photography, FAF, and OCT findings. Based on the foveal appearance on B-scan SD-OCT, patients’ eyes were categorized into two groups: Eyes with a fovea plana appearance (‘FP group’) and eyes without a fovea plana appearance (‘no FP group’).

### 2.3. B-Scan SD-OCT Analysis

SD-OCT examinations included B-scan macular line and volume scans. The following SD-OCT characteristics were analyzed: (1) Fovea plana appearance on OCT B-scans was defined as the persistence of IRL at the fovea, with at least the outer plexiform layer (OPL). Grading of the fovea plana was established based on the previously described Thomas grading classification of foveal hypoplasia using SD-OCT (grade 1 to 4 and atypical) [18]. (2) Subfoveal choroidal thickness (SFCT) was manually measured using the caliper of the Heidelberg software between the hyperreflective band corresponding to the RPE/Bruch’s membrane complex and the outer hyperreflective band at the choroidal–scleral interface. Measurement was made on an enhanced-depth imaging (EDI) B-scan when available, or on a good-enough-quality standard B-scan OCT when the EDI B-scan was unavailable. Finally, (3) “pachyvessels” were defined as the presence of large vessels with hyperreflective walls in Haller’s layer, underlying a thinned choriocapillaris, in the macular region [19].

### 2.4. OCT-A Analysis

OCT-A imaging was generated via the software for Spectralis OCT-A (version 1.10.4.0), using a 15° × 15° field of view around the macular region. OCT-A images were assessed for the presence of bridging vessels in the FAZ. As the presence of vitelliform material disturbs detailed segmentation, slabs containing all superficial, intermediate, and deep capillary plexuses together were used. The boundaries for the retinal capillary plexuses were defined from the inner limiting membrane (ILM) to OPL. When FAZ integrity was preserved, its corresponding area was manually measured in mm^2^ on slabs ranging from the ILM to the OPL. Double-masked measurements were performed with a third measurement in the case of a difference of more than 10% between both.

### 2.5. Statistical Analysis

Data were collected and statistically analyzed using R (version 4.0.3). Continuous variables are presented as mean ± standard deviation (SD), and categorical variables as frequency and percentage. The *T*-test (or Mann–Whitney U test when necessary) was used to compare continuous variables between groups, whereas the Chi-squared test (or Fisher’s exact test when necessary) was used to compare qualitative parameters. When more than 2 groups were analyzed, an analysis of variance (ANOVA) test (or Kruskal–Wallis test when necessary) was used for continuous variables. Significance was evaluated at the level of *p* < 0.05.

## 3. Results

### 3.1. Patients Characteristics

Sixty-six eyes of thirty-three patients (16 females (48.5%) and 17 males (51.5%)) with Best disease were analyzed. Overall, the mean age of subjects was 31.2 ± 19.0 years old. The diagnosis of Best disease was established based on genetic confirmation for 13 patients (39.4%), an EOG Arden ratio less than 170% for 4 patients (12.1%), clinical presentation and an EOG Arden ratio between 170 and 179% for 1 patient (3.0%), family history and clinical presentation for 10 patients (30.3%), and finally, clinical presentation alone for 5 patients (15.2%) (Table 1). A total of seven eyes of six patients could not be evaluated because of the poor quality of OCT images and were excluded from the study. Therefore, only 59 eyes of 32 patients were analyzed (Figure 1). Based on the foveal appearance on B-scan SD-OCT, 16 eyes (27.1%) of 9 patients had a fovea plana appearance (‘FP group’) (representative case, Figure 2) and 43 eyes (72.9%) of 23 patients did not have fovea plana appearance (‘no FP group’). The mean age of subjects was statistically lower in the ‘FP group’ compared to the ‘no FP group’ [19.2 ± 14.1 and 35.9 ± 18.9, respectively (*p* = 0.01)]. Otherwise, medical records and demographic criteria were all equal between both groups (Table 2 and Table 3). Using the Thomas classification (14), the grading of foveal hypoplasia was atypical for 14 eyes (87.5%) and grade 1b for 2 eyes (12.5%) (Table 1).

### 3.2. B-Scan SD-OCT Findings

There was no significant difference between both groups with regard to mean choroidal thickness [338.3 ± 92.2 and 360.5 ± 102.7, respectively (*p* = 0.7)] and the presence of pachyvessels [12 (75.0%) and 35 (81.4%), respectively (*p* = 0.7)] (Table 3). Moreover, these two OCT parameters did not statistically differ along the stage of the disease (Table 4). Finally, the presence of fovea plana was neither related to the stage of the disease nor to the persistence of vitelliform material under the macula (Table 3).

### 3.3. OCT-A Findings

Out of the 16 eyes with fovea plana appearance on B-scan SD-OCT, 13 eyes had an interpretable OCT-A, which revealed bridging vessels passing through the FAZ in all eyes (representative case, Figure 3) (Table 5). On the other hand, 27 eyes out of the 43 eyes (62.8%) with no fovea plana appearance on B-scan SD-OCT had analyzable OCT-A, with a FAZ identified and measured in all eyes (Table 5). The mean FAZ area was 0.34 ± 0.17 mm^2^, which was independent of the disease stage (Table 6).

## 4. Discussion

With its specific software algorithm based on split-spectrum amplitude decorrelation angiography, OCT-A allowed a noninvasive visualization of both retinal and choroidal microvascular flow and structure at various depths in different macular and choroidal pathologies, with a precise anatomical location. Until recently, only one paper by Mirshahi et al. described the presence of intriguing bridging vessels pattern across the FAZ, a capillary plexus in an area that is mostly considered avascular (11). In our study, 16 eyes of 9 patients (27.1%) with Best disease had a fovea plana appearance with the persistence of INL on B-scan SD-OCT, out of which 13 (81%) could be analyzed with OCT-A images. We identified bridging vessels passing through the FAZ in the retinal capillary plexuses in all of these analyzable eyes. In contrast, none of the OCT-A images of patients with no fovea plana appearance on B-scan SD-OCT showed bridging vessels across the FAZ. As the use of OCT-A is of high accuracy in diagnosing FP in healthy retinas [13], the concordance between OCT and OCT-A findings in our cohort gives a good value to evocate an association between FP and Best disease. Even if the incidence of the fovea plana in the general population with cataracts is approximately 9% [20], the incidence in this cohort seems much higher.

Pathophysiology of fovea plana is still widely debated. It is supposed to be due to foveal misdevelopment during embryogenesis. Normal development of the human fovea begins during midgestation (11 weeks of gestation) and is completed 15–45 months after birth. In the early stage of development, all of the retinal layers are present at the fovea, with only one layer of cones. Then, the fovea progressively matures in three distinct steps: (1) The centrifugal displacement of the IRL to form the foveal depression; (2) the centralized migration of the cones; and (3) the increase in cones’ number and density, with thinning and elongation of the cones’ OS, and photoreceptors’ axons elongation at the level of the OPL [14,21,22,23].

Various degrees of fovea plana are described. The structural grading system of foveal hypoplasia using SD-OCT images proposed by Thomas et al. [14,18] gives an indication of the stage at which foveal development was arrested (grades 1 to 4). Furthermore, this classification includes the ‘atypical’ grade, which is characterized by a shallower foveal pit with disruption of the inner ellipsoid zone.

All but two of the eyes with FP in our cohort had atypical foveal hypoplasia. Patient 18 is the only patient that had grade 1b foveal hypoplasia in both eyes, while she presented none of the clinical features of Best disease (Figure 3). Diagnosis of BVMD was established based on familial history and genetic testing in this young woman. This particular case supports the hypothesis that the mechanism of the fovea plana is neither dependent on the deposition of the material nor the stage of the disease. Considering this, we believe that the presence of foveal hypoplasia should increase the suspicion of Best disease in children with a positive familial history of BVMD, even before the appearance of vitelliform material deposition. Furthermore, our observations in patient 18 suggest that the ‘atypical’ aspect of foveal hypoplasia might appear as the disease progresses, with secondarily acquired disruption of external retinal layers.

Atypical foveal hypoplasia has been described in achromatopsia and cone-rod dystrophy, particularly in patients with a mutation of the encoding activating transcription factor 6A (ATF6) gene [24,25]. ATF6 encodes a ubiquitous endoplasmic reticulum (ER) stress-regulated transmembrane transcription factor, required for ER stress-response and transcriptional induction from ER stress-response elements. In patients with ATF6 mutation, ER stress is increased. ATF6 is also essential for human cone development [26]. The exact pathomechanism of fovea plana in ATF6-muted patients is unclear, but it is supposed that the ATF6 gene plays an important role in the development and function of the fovea and cone photoreceptors [24], possibly through the increased ER stress.

Milenkovic et al. showed that autosomal recessive BEST1 mutants have increased susceptibility to ER stress. Although this has not been demonstrated for dominant BEST1 mutants [27], and based on the hypothetical link between increased ER stress and foveal development/atypical hypoplasia in achromatopsia secondary to ATF6 mutation, we hypothesize that ER stress might also play a role in the development of fovea plana in Best disease.

Three out of our patients with FP had genetic confirmation of BEST1 mutation. As the three of these mutations were different, we hypothesize that FP appearance in Best disease is not specific to one mutation.

The choroid was thick in many patients, and we encountered several pachyvessels. The significance of choroidal thickness in Best disease and the associated pachyvessels need more studies to be confirmed and understood [28].

Our results showed that patients with no FP appearance had significantly higher FAZ areas than the normal population. Indeed, we measured a mean FAZ area of 0.34 ± 0.17 mm^2^, which seems to be higher than the FAZ area measured in healthy subjects in the literature (0.266 ± 0.097 mm^2^) [29]. This is also consistent with previous studies that have shown enlarged FAZ at the deep capillary plexuses and demonstrated vascular abnormalities at both superficial and deep capillary plexuses in patients with Best disease [10,30]. These anomalies were described as possibly due to the pressure of the vitelliform lesion that induces capillary dropout and patchy vascularity loss, or to neurovascular degeneration, as a result of the reduced trophic support from target plexuses or from broader disruption of retinal architecture secondary to the expansion of the vitelliform lesion. As the FAZ area did not statistically differ along with the stage of the disease in our cohort, we suppose that the presence of the material and the potential vascular abnormalities did not influence the size of the FAZ. Finally, we believe that the vascular endothelial growth factor (VEGF) could play an essential role in the underlying mechanism of this microvascular process. In fact, pathologic RPE changes and retinal vascular diseases that involve FAZ, as seen in Best disease, could stimulate the production of VEGF, which may play a crucial role in inducing vasculogenesis in usually avascular areas of FAZ [11].

Our study has several limitations. First, it includes a relatively small sample of patients, which is due to the rarity of the disease. Second, its retrospective and monocentric design does not necessarily extrapolate the data to other units and the general population. Finally, missing data regarding genetic testing and OCT-A images may have biased our results. Therefore, further investigations with larger prospective studies remain crucial to fully validate our results.

## 5. Conclusions

In conclusion, the present study showed that foveal hypoplasia is found in a significant number of patients with Best disease. BMVD should be highly suspected in children with foveal hypoplasia when there is a positive familial history of the disease, even before the appearance of the characteristic vitelliform deposition. For the patients that have no FP appearance, the FAZ area seems to be enlarged compared to the normal population.

## Figures and Tables

**Figure 1 jpm-13-00337-f001:**
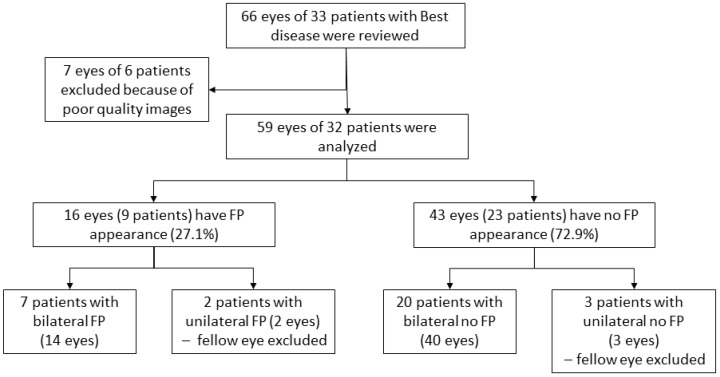
Flow-chart of included patients with Best disease, divided by the aspect of the fovea (fovea plana (FP) or no FP).

**Figure 2 jpm-13-00337-f002:**
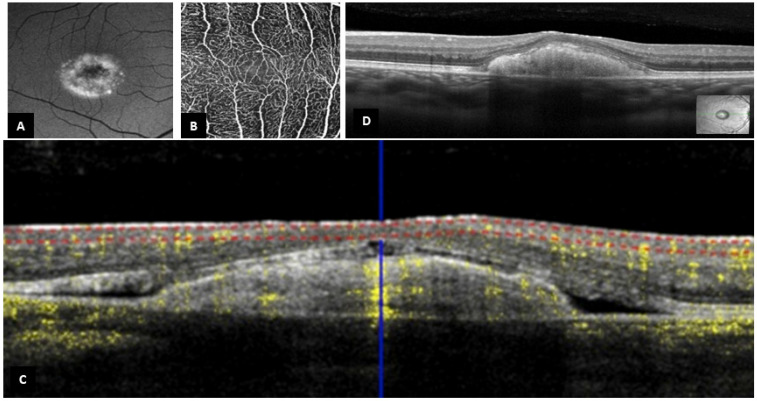
Representative case (Patient 2 right eye). (**A**). Autofluorescence of the right eye showing the intense hyperautofluorescence of the material under the macula. (**B**). OCT-A of the right eye with bridging vessels across the foveal avascular zone (the slabs contain the superficial and the deep capillary plexuses together). (**C**). Corresponding OCT-A B-scan of the right eye showing the decorrelation signal in inner retinal layers. (**D**). Central OCT B-scan from the same patient, with persistence of the outer plexiform layer.

**Figure 3 jpm-13-00337-f003:**
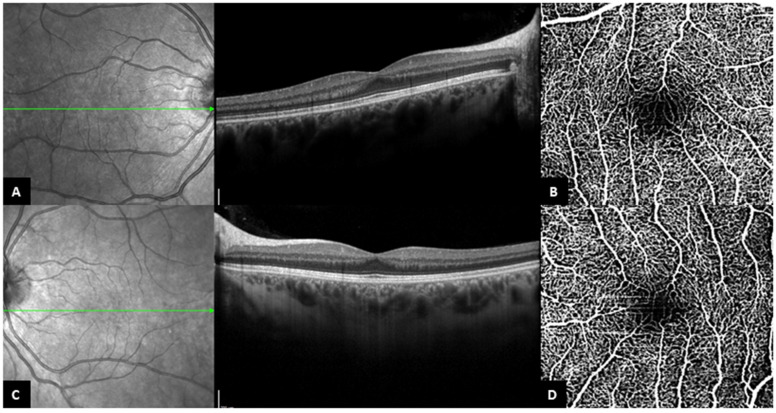
Patient 18. (**A**,**C**). Central OCT B-scan of the right eye and the left eye, with persistence of the inner retinal layers. (**B**,**D**). OCT-A from the same patient, ((**B**). right eye, (**D**). left eye), with bridging vessels across the foveal avascular zone (the slabs contain the superficial and deep capillary plexuses together).

**Table 1 jpm-13-00337-t001:** Patient’s details and characteristics.

Patient	Age (yrs)	Sex	R/L	SE (D)	VA (logMAR)	IOP	FP Y/N/E	FP Grade	Diagnose Made Based on	BEST1 Mutation
1	7	F	R L	- -	0.1 0.1	- -	Y E	Atypical -	Familial history and clinical examination	-
2 FP	7	F	R L	+1.875 +1.625	0.0 0.1	16 16	Y Y	Atypical Atypical	Familial history, clinical examination, EOG Arden ration less than 170%, genetic confirmation	het. mutation c.44G>A p.(Gly15Asp) exon 2
3	26	F	R L	+3.25 +1.50	0.7 0.18	14 15	N N	- -	Clinical examination, EOG Arden ration less than 170%, genetic confirmation	mutation het.c.44G>A, p.(Gly15Asp) exon 2
4	78	F	R L	- -	0.9 0.9	11 12	N N	- -	Familial history and clinical examination	-
5	55	M	R L	+4.0 +3.625	0.152 0.3	14 14	N N	- -	Familial history, clinical examination, EOG Arden ration less than 170%	-
6	35	M	R L	+0.125 +0.625	0.0 0.0	11 9	N N	- -	Familial history, clinical examination, genetic confirmation	het. mutation c.905A>C (Asp302Ala)
7	15	M	R L	+0.125 +1.0	−0.10 0.0	17 18	Y Y	Atypical Atypical	Clinical examination alone	-
8	34	F	R L	−1.625 −1.625	0.3 0.3	11 14	N N	- -	Clinical examination alone	-
9	62	F	R L	- -	0.045 0.045	16 14	N N	- -	Familial history, clinical examination, EOG Arden ration 170–179%, genetic confirmation	het. mutation c.97T>C, p.(Tyr33His) exon 2
10	37	F	R L	+0.125 +0.125	0.4 0.2	18 13	N N	- -	Clinical examination alone	-
11	57	F	R L	+3.0 +2.325	0.3 0.6	15 13	E E	- -	Familial history, clinical examination, EOG Arden ration less than 170%	-
12	29	F	R L	−0.50 −2.25	0.3 0.3	14 17	N N	- -	Familial history, clinical examination, EOG Arden ration less than 170%, genetic confirmation	het. mutation c.916G>C, p.(Glu306Gln) exon 8
13	11	F	R L	+1.375 +1.625	0.0 0.0	16 16	N N	- -	Clinical examination, EOG Arden ration 170–179%	-
14	45	M	R L	+1.125 +1.75	0.3 0.2	13 15	N N	- -	Familial history, clinical examination, EOG Arden ration less than 170%, genetic confirmation	het. mutation c.10A>G (p.Thr4Ala) exon 2
15	6	F	R L	+5.50 +0.125	0.152 0.7	17 17	Y E	Atypical -	Clinical examination and genetic confirmation	hom. mutation (recessive) IVS5+ 1G>A
16	45	F	R L	- -	0.2 0.2	13 14	N N	- -	Clinical examination, EOG Arden ration less than 170%	-
17	58	M	R L	+5.625 +5.75	0.3 1.0	18 18	N N	- -	Familial history, clinical examination, genetic confirmation	het. mutation c.692G>C, p.(Ser231Thr) exon 6
18	29	F	R L	−0.125 −0.625	0.0 0.0	13 15	Y Y	1b 1b	Familial history and genetic confirmation	het. mutation c.692G>C, p.(Ser231Thr) exon 6
19	19	F	R L	+2.125 +1.375	0.3 0.5	15 16	N N	- -	Familial history and clinical examination	-
20	11	M	R L	+1.625 +1.50	0.3 0.4	12 15	Y Y	Atypical Atypical	Familial history and clinical examination	-
21	15	M	R L	- -	0.4 0.7	- -	E N	- -	Familial history, clinical examination, genetic confirmation	het. mutation c.25G>T, p.(Val9Leu) exon 2
22	12	M	R L	+0.25 +3.75	0.5 0.0	17 18	N N	- -	Familial history, clinical examination, genetic confirmation	het. mutation c.25G>T, p.(Val9Leu) exon 2
23	10	F	R L	+0.50 +0.625	0.0 0.0	- -	N N	- -	Familial history and clinical examination	-
24	14	M	R L	- -	0.0 0.1	- -	N N	- -	Familial history and clinical examination	-
25	37	M	R L	- -	0.4 0.1	- -	E N	- -	Familial history, clinical examination, genetic confirmation	het. mutation c.217A>T, p.(Ile73Phe) exon 3
26	37	M	R L	+6.0 +4.625	1.30 0.2	13 16	E N	- -	Familial history, clinical examination, EOG Arden ration less than 170%, genetic confirmation	het. mutation c.916G>C, p.(Glu306Gln) exon 8
27	14	M	R L	−0.125 +1.25	0.4 0.0	20 19	Y Y	Atypical Atypical	Familial history and clinical examination	-
28	16	M	R L	+5.50 +6.0	0.0 0.6	20 21	N N	- -	Familial history and clinical examination	-
29	49	M	R L	+4.75 +5.375	0.20 1.1	19 20	N N	- -	Familial history and clinical examination	-
30	47	M	R L	+0.25 +0.125	0.5 0.7	12 12	Y Y	Atypical Atypical	Clinical examination alone	-
31	31	M	R L	+0.125 0	0.0 0.0	12 15	N N	- -	Clinical examination and EOG Arden ration less than 170%	-
32	60	F	R L	- -	−0.10 1.1	14 15	N N	- -	Familial history and clinical examination	-
33	31	M	R L	−2.25 −2.25	0.4 0.0	17 18	Y Y	Atypical Atypical	Clinical examination alone	-

F: Female, M: Male; R: Right, L: Left; SE: Phakic spherical equivalent; VA: Visual acuity; FP: Fovea plana; Y: Yes, N: No, E: Excluded eye, -: Data missing or unavailable; het.: Heterozygote, hom.: Homozygote.

**Table 2 jpm-13-00337-t002:** Demographic characteristics of patients, according to the appearance of the fovea.

	FP Group (n = 9)	No FP Group (n = 23)	All (n = 32)	*p*-Value
Gender Male (%) Female (%)	5 (55.6%) 4 (44.4%)	12 (52.2%) 11 (47.8%)	17 (53.1%) 15 (46.9%)	NA
Age (years) Mean (SD) Median (Q1, Q3) Min-Max	19.17 (14.10) 14.70 (7.40, 29.80) 6.70–47.30	35.92 (18.87) 35.40 (18.20, 47.45) 10.10–78.10	31.21 (19.04) 30.85 (14.57, 45.32) 6.70–78.10	0.01 †

† *T*-test.

**Table 3 jpm-13-00337-t003:** Clinical and OCT characteristics of patients, according to the appearance of the fovea.

	FP Group (n = 16)	No FP Group (n = 43)	All (n = 59)	*p*-Value
Eyes RE (%) LE (%)	9 (56.2%) 7 (43.8%)	20 (46.5%) 23 (53.5%)	29 (49.2%) 30 (50.8%)	0.5 *
Visual acuity (logMAR) Mean (SD) Median (Q1, Q3) Min-Max	0.18 (0.23) 0.10 (0.00, 0.40) −0.10–0.70	0.29 (0.33) 0.20 (0.00, 0.35) −0.10–1.10	0.26 (0.30) 0.20 (0.00, 0.40) −0.10–1.10	0.3 †
IOP Mean (SD) Median (Q1, Q3) Min-Max Missing	15.80 (2.60) 16.00 (14.00, 17.50) 12.00–20.00 1	15.08 (2.75) 15.00 (14.00, 17.00) 9.00–21.00 6	15.29 (2.70) 15.00 (13.75, 17.00) 9.00–21.00 7	0.4 ‡
Spherical equivalent (phakic eyes only) (D) Mean (SD) Median (Q1, Q3) Min-Max Missing	0.63 (1.86) 0.25 (−0.12, 1.56) −2.25–5.50 1	1.92 (2.40) 1.38 (0.12, 3.88) −2.25–6.00 3	1.50 (2.30) 1.19 (0.12, 2.97) −2.25–6.00 4	0.05 ‡
Stage of the disease 1 2 3 4 5 6	2 (12.5%) 0 (0.0%) 3 (18.8%) 3 (18.8%) 0 (0.0%) 8 (50.0%)	3 (7.0%) 6 (14.0%) 7 (16.3%) 2 (4.7%) 4 (9.3%) 21 (48.8%)	5 (8.5%) 6 (10.2%) 10 (16.9%) 5 (8.5%) 4 (6.8%) 29 (49.2%)	0.3 **
Persistence of vitelliform material under the macula No (%) Yes (%)	2 (12.5%) 14 (87.5%)	8 (18.6%) 35 (81.4%)	10 (16.9%) 49 (83.1%)	0.7 **
Choroidal thickness (µm) Mean (SD) Median (Q1, Q3) Min-Max Missing	338.25 (92.20) 360.00 (306.50, 395.50) 92.00–445.00 0	360.52 (102.71) 364.50 (319.75, 404.00) 118.00–613.00 3	354.16 (99.51) 364.50 (311.75, 398.50) 92.00–613.00 3	0.7 †
Presence of pachyvessels No (%) Yes (%)	4 (25.0%) 12 (75.0%)	8 (18.6%) 35 (81.4%)	12 (20.3%) 47 (79.7%)	0.7 **

* Chi-squared test. † Mann–Whitney U test. ‡ *T*-test. ** Fisher’s exact test.

**Table 4 jpm-13-00337-t004:** Choroidal thickness and pachyvessels according to the stage of the disease.

	Stage I (n = 5)	Stage II (n = 6)	Stage III (n = 10)	Stage IV (n = 5)	Stage V (n = 4)	Stage VI (n = 29)	*p*-Value
Choroidal thickness (µm)							0.7 *
Mean (SD)	374.20 (54.73)	394.40 (88.62)	350.00 (40.49)	353.00 (85.85)	302.25 (74.08)	352.44 (127.01)	
Median (Q1, Q3)	394.00 (352.00, 397.00)	361.00 (339.00, 435.00)	334.50 (320.25, 385.50)	397.00 (292.00, 417.00)	313.00 (258.25, 357.00)	368.00 (311.50, 401.50)	
Min-Max	292.00–436.00 0	308.00–529.00	293.00–407.00	233.00–426.00	211.00–372.00	92.00–613.00	
Missing	374.20 (54.73)	1		0	0	2	
Pachyvessels							0.4 †
No (%)	1 (20.0%)	1 (16.7%)	2 (20.0%)	2 (40.0%)	2 (50.0%)	4 (13.8%)	
Yes (%)	4 (80.0%)	5 (83.3%)	8 (80.0%)	3 (60.0%)	2 (50.0%)	25 (86.2%)	

* Kruskal Wallis test. † Fisher’s exact test.

**Table 5 jpm-13-00337-t005:** OCT-A findings.

	FP Group (n = 13)	No FP Group (n = 27)	All (n = 40)	*p*-Value
Bridging vessels across de fovea				<0.001 *
No (%)	0 (0%)	27 (100%)	27 (67.5%)	
Yes (%)	13 (100%)	0 (0%)	13 (32.5%)	
FAZ area (mm^2^)		0.34 ± 0.17 mm^2^		

* Fisher’s exact test.

**Table 6 jpm-13-00337-t006:** FAZ area according to the stage of the disease.

	Stage I (n = 5)	Stage II (n = 6)	Stage III (n = 10)	Stage IV (n = 5)	Stage V (n = 4)	Stage VI (n = 29)	*p*-Value
FAZ area (mm^2^)							0.2 *
Mean (SD)	0.41 (0.01)	0.29 (0.03)	0.34 (0.14)	0.20 (0.13)	0.45 (0.07)	0.36 (0.23)	
Median (Q1, Q3)	0.41 (0.41, 0.42)	0.29 (0.28, 0.31)	0.38 (0.31, 0.44)	0.20 (0.15, 0.24)	0.45 (0.42, 0.47)	0.30 (0.24, 0.36)	
Min-Max	0.41–0.42	0.25–0.33	0.10–0.46	0.11–0.29	0.40–0.50	0.19–1.00	
Missing	3	2	4	3	2	18	

* Kruskal Wallis test.

## Data Availability

Data is unavailable due to privacy or ethical restriction.

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
