# Peer review of "Atypical Foveal Hypoplasia in Best Disease"

_jpm, 2023, doi:10.3390/jpm13020337_

Round 1
Reviewer 1 Report
This manuscript tried to investigate the prevalence and characteristics of foveal hypoplasia in patients with Best Disease by employing SD-OCT/OCTA. Overall, I think the concept behind the study is of interest. In the present study, 59 eyes of 32 patients with Best disease were included. The authors found that foveal hypoplasia was present in 27.1% of patients with Best disease. Interestingly, bridging vessels were seen through the FAZ in all the eyes, which may provide a novel strategy in the early diagnosis and treatment of Best Disease.
Specifically, the authors should fix the following problems which will help the work achieve the published criteria of this journal.
1. Please go through the text to check the sample size. There are many mistakes in counting the numbers. In Page 1, lines 15-16, “Fifty-nine eyes of 32 patients (11 females (47.8%) and 12 males (52.2%), P=0.9”. The percentage of the female and male is wrong. And please check the number in Results Section.
2. Additionally, in Page 1, line 24, it is mentioned that “OCT-A done in 14 eyes showed bridging vessels through the FAZ in 100% of eyes.” However, in page 9, lines 203-205, “Out of the 16 eyes with fovea plana appearance on B-scan SD-OCT, 13 eyes had an interpretable OCT-A, which revealed bridging vessels passing through the FAZ in all eyes 204 (representative case, Figure 2) (Table 5). ” The numbers are inconsistent. Please check carefully and explain.
3. By the way, only 14 eyes (maybe, should be checked) out of 16 eyes showed bridging vessels in FP group, it is incorrect to indicate that the bridging vessels through the FA in all eyes.
4. Figure 3 should be included in the text in page 9, line 205” (representative case, Figure 2) (Table 5)”.
5. Is the FP associated with the stage of the disease?
6. How to distinguish the FAZ integrity in all the cases, especially before bridging vessels through the FAZ area? How to ensure the accuracy of the FAZ measurement?
7. How many layers were the bridge vessels across the FAZ area? Is it associated with the volume of sub-macula?
8. It should be more clinical significance to analyze the vessel density and fractal dimension, and tortuousity of the bridging vessels.
9. It is much better to have clinical statistician to check the data and the statistical methods.

Author Response
Thank you for your comments. See our answers in blue following your comments.
This manuscript tried to investigate the prevalence and characteristics of foveal hypoplasia in patients with Best Disease by employing SD-OCT/OCTA. Overall, I think the concept behind the study is of interest. In the present study, 59 eyes of 32 patients with Best disease were included. The authors found that foveal hypoplasia was present in 27.1% of patients with Best disease. Interestingly, bridging vessels were seen through the FAZ in all the eyes, which may provide a novel strategy in the early diagnosis and treatment of Best Disease.
Specifically, the authors should fix the following problems which will help the work achieve the published criteria of this journal.
- Please go through the text to check the sample size. There are many mistakes in counting the numbers. In Page 1, lines 15-16, “Fifty-nine eyes of 32 patients (11 females (47.8%) and 12 males (52.2%), P=0.9”. The percentage of the female and male is wrong. And please check the number in Results Section.
We are sorry about this mistake, we modified the amount and percentage which were correctly mentioned in the table.
- Additionally, in Page 1, line 24, it is mentioned that “OCT-A done in 14 eyes showed bridging vessels through the FAZ in 100% of eyes.” However, in page 9, lines 203-205, “Out of the 16 eyes with fovea plana appearance on B-scan SD-OCT, 13 eyes had an interpretable OCT-A, which revealed bridging vessels passing through the FAZ in all eyes 204 (representative case, Figure 2) (Table 5). ” The numbers are inconsistent. Please check carefully and explain.
Thank you for noticing this. There has been a mistake and it was 13 patients as mentioned in the table 5.
- By the way, only 14 eyes (maybe, should be checked) out of 16 eyes showed bridging vessels in FP group, it is incorrect to indicate that the bridging vessels through the FA in all eyes.
From the 13 patients with FP, all had bridging vessel, the other patients with FP had no OCT-A interpretable. This is why we mentioned the 100%. We rementionned “with OCT-A” in the sentence to make it clear.
- Figure 3 should be included in the text in page 9, line 205” (representative case, Figure 2) (Table 5)”.
Thank you for this comment, figure 3 is now mentioned page 9.
- Is the FP associated with the stage of the disease?
FP was not associated either with FAZ or stage of the disease. We mentioned it in the result “independently with the stage of the disease” (line 168).
- How to distinguish the FAZ integrity in all the cases, especially before bridging vessels through the FAZ area? How to ensure the accuracy of the FAZ measurement?
The FAZ was analyzed combining all the layers because of the subretinal material as mentioned in the methods. The accuracy of the FAZ measurement was guarantied with :“Double masked measurement were performed with a third measurement in case of difference more than 10% between both. “ (page 4, line 144)
- How many layers were the bridge vessels across the FAZ area? Is it associated with the volume of sub-macula?
It is very difficult to answer this question. The thinness of the retina didn’t enable such a precision. Our feeling is that it was in all three layers.
- It should be more clinical significance to analyze the vessel density and fractal dimension, and tortuousity of the bridging vessels.
Indeed we are sorry that we could not perform those analyses. The spectralis software is not complete yet and we had not the tool to perform those analyses. This would have been very relevant.
- It is much better to have clinical statistician to check the data and the statistical methods.
The statistical analyses have been performed by our statistical department using R. We are sorry about the number mistakes.
Reviewer 2 Report
This is a nice project studying FP in Best disease. Data are clearly presented. The reviewer is impressed with the number of the patients with such a rare disease.
A concern for this study is the significance of data of choroidal thickness (CT) and pachyvessel. Pachychoroid is a hallmark of central serous chorioretinopathy. In discussion, the group did not mention the significance of CT in Best disease. The reviewer feels that data and references about CT are not needed in this paper.
Author Response
This is a nice project studying FP in Best disease. Data are clearly presented. The reviewer is impressed with the number of the patients with such a rare disease.
Thank you for that comment
A concern for this study is the significance of data of choroidal thickness (CT) and pachyvessel. Pachychoroid is a hallmark of central serous chorioretinopathy. In discussion, the group did not mention the significance of CT in Best disease. The reviewer feels that data and references about CT are not needed in this paper.
Indeed this is not the main result of our cohort. However, this finding might be interesting for later studies on choroid since there is very few data on this subject with this amount of patients. We mentioned the data on choroid in the discussion, mentioning the need for more studies. “The choroid was thick in many patients, and we encountered several pachyvessels. The significance of choroidal thickness in Best disease and the associated pachyvessels need more studies to be confirmed and understood.” (page 13, line 292)
Round 2
Reviewer 1 Report
The authors did not check carefully through the text. For example, in the first sentence of the result section, ''Sixty six eyes of 33 patients 11 females (47.8%) and 12 males (52.2%), P=0.9 with Best 158 disease were analyzed." The number is obviously wrong.
Author Response
We are sorry about this mistake, it has been changed in the manuscript and reviewed. There were 16 females and 17 males before exclusion.